# Gibberellin Oxidase Gene Family in *L. chinense*: Genome-Wide Identification and Gene Expression Analysis

**DOI:** 10.3390/ijms22137167

**Published:** 2021-07-02

**Authors:** Lingfeng Hu, Pengkai Wang, Zhaodong Hao, Ye Lu, Guoxia Xue, Zijian Cao, Haoxian Qu, Tielong Cheng, Jisen Shi, Jinhui Chen

**Affiliations:** 1Key Laboratory of Forest Genetics and Biotechnology of Ministry of Education, Co-Innovation Center for Sustainable Forestry in Southern China, Nanjing Forestry University, Nanjing 210037, China; hlf625@nifu.edu.cn (L.H.); haozd1992@163.com (Z.H.); luye@njfu.edu.cn (Y.L.); xueguoxia@njfu.edu.cn (G.X.); caozijianv@163.com (Z.C.); quhaoxian@163.com (H.Q.); chengtl@njfu.edu.cn (T.C.); jshi@njfu.edu.cn (J.S.); 2College of Horticulture Technology, Suzhou Polytechnic Institute of Agriculture, Suzhou 215000, China; mengxiao02181@hotmail.com

**Keywords:** gibberellin oxidase, *L. chinense*, gene family, abiotic stress, phytohormone

## Abstract

GAox is a key enzyme for the transformation of gibberellins, and belongs to the 2-ketoglutarate dependent dioxygenase gene family (2ODD). However, a systematic analysis of *GAox* in the angiosperm *L. chinense* has not yet been reported. Here, we identified all *LcGAox* gene family members in *L. chinense*, which were classified into the three subgroups of *GA20ox*, *C19GA2ox*, and *C20GA2ox*. Comparison of the gene structure, conserve motifs, phylogenetic relationships, and syntenic relationships of gibberellin oxidase gene families in different species indicated that the gene functional differences may be due to the partial deletion of their domains during evolution. Furthermore, evidence for purifying selection was detected between orthologous *GAox* genes in rice, grape, Arabidopsis, and *L. chinense*. Analysis of the codon usage patterns showed that mutation pressure and natural selection might have induced codon usage bias in angiosperms; however, the *LcGAox* genes in mosses, lycophytes, and ambarella plants exhibited no obvious codon usage preference. These results suggested that the gibberellin oxidase genes were more primitive. The gene expression pattern was analyzed in different organs subjected to multiple abiotic stresses, including GA, abscisic acid (ABA), and chlormequat (CCC) treatment, by RNA-seq and qRT-PCR, and the stress- and phytohormone-responsive cis-elements were counted. The results showed that the synthesis and decomposition of GA were regulated by different *LcGAox* genes in the vegetative and reproductive organs of *L. chinense*, and only *LcGA2ox1,4,* and *7* responded to the NaCl, polyethylene glycol, 4 °C, GA, ABA, and CCC treatment in the roots, stems, and leaves of seedlings at different time periods, revealing the potential role of *LcGAox* in stress resistance.

## 1. Introduction

As a class of tetracyclic diterpenes, gibberellins (GA) are indispensable for the growth and development of plants. Some bioactive GAs are widely involved in various biological activities [1], such as seed germination [2], stem and shoot elongation [3,4], leaf expansion [5], and late flower and fruit development [6,7]. The regulatory synthesis pathway of GA has been studied in depth in model plants, and the related regulatory genes have been successively confirmed by scientists [8]. The GA synthesis pathway can be divided into three main steps: the first step is the catalysis of geranylgeranyl pyrophosphat (GGPP) from ent-kaurene by the enzymes of CPS and KS in the piroplasmid; the second step is the synthesis of GA12 according to KO and KAO enzymes by the products of the previous step in the endomembrane; and the third step is the synthesis of physiologically active GA under the catalysis of the gibberellin oxidase (*GAox*) enzyme [9,10,11]. 

Liriodendron is an ancient relict with only two species in the genus in nature, including *L. chinense* and *L. tulipifera*. Due to their excellent material properties and ornamental value, they are widely planted in China and the United States for landscaping and wood production [12]. The GA signaling pathway plays an important role in promoting plant height and regulating flowering [13]. *GAox* belongs to the 2OG-Fe (II) oxygenase superfamily, which is divided into multiple small families based on structural and functional differences [11]. The *GA20ox*, *GA2ox*, and *GA3ox* gene families play an important role in GA biosynthesis and degradation [14]. In the last stage of GA biosynthesis, GA 20-oxidase (*GA20ox*) and GA 3-oxidase (*GA3ox*) are considered to be the key enzymes in the pathways that convert GA12 and GA53 into GA4 and GA1, respectively [6,11]. Increasing or decreasing *GA20ox* expression levels can improve the fruit setting number and biomass of different species [7,15,16]. The loss-of-function of *GA20ox* in rice was expressed in the shoots but not the reproductive organs, leading to increased grain yields [17]. *GA3ox* controls plant growth and development mainly by regulating endogenous gibberellin homeostasis [18,19,20,21,22]. *GA2ox* contains the two subgroups *C19GA2ox* and *C20GA2ox*, which are relatively independent in terms of evolution [9]. In tomato, seed germination and development, fruit weight and firmness, and branch growth are all related to *GA2ox* [7,23,24]. Overexpressing *GA2ox* in *Arabidopsis* and tobacco could confer dwarfism [25]. The GA 2-oxidase (*GA2ox*) can preferentially act on physiologically active GA or their inactive precursors to achieve GA degradation [7]. The regulation of *GAox* expression has also been explored in multiple species. Research has found that *GAox* plays an important role not only in development and growth but also in the response to stimulation by different exogenous hormones and abiotic stresses [26,27]. *OsGA2ox5*-OE and *AtGA2ox7*-OE plants demonstrate tolerance to high-salt treatment [28]. The seedlings of the *ga20ox2-1* mutant exhibited primary root and root hair elongation in comparison with the wild-type under NaCl treatment [26]. Under the influence of GA or GA biosynthesis inhibitors, *GAox* shows different degrees of functional differentiation and redundancy in cucumbers, grapes, and other species [14,29].

Understanding the GA metabolic regulation pathway in *L. chinense* can provide a molecular basis for the targeted breeding of new varieties. Gibberellin oxidase is considered to be an important catalyst and synthase responsible for the interconversion of different GAs in the final stage of gibberellin biosynthesis pathway. However, there appear to be no related reports on *L. chinense*. In this study, we identified 11 *GAox* gene belonging to the *GA20ox* and *GA2ox* bases in the *L. chinense* genomic database. The *LcGAox* genes were systematically analyzed in terms of basic physical and chemical properties, gene structure, conserved domains, and selective evolutionary relationships. The evolutionary status of *LcGAox* was explored via codon usage pattern analysis of *GAox* genes in 21 species. RNA sequencing (RNA-seq) and quantitative real-time PCR (qRT-PCR) were used to analyze the *LcGAox* gene expression patterns in multiple organs and under different abiotic stresses, as well as under phytohormone stimulation. This study provides research basis for further understanding the biological function of gibberellin oxidase in *L. chinense.*

## 2. Result

### 2.1. Identification of GA2ox and GA20ox genes in L. chinense

The *GA20ox*, *GA2ox*, and *GA3ox* genes belong to the 2–oxoglutarate-dependent dioxygenase superfamily in plants and play an important role in the final step of GA biosynthesis. The *GAox* members of *O. sativa*, *A. thaliana*, *A. trichopoda*, and *V. vinifera* are listed in Table 1. Through BLASTP and HMMER analysis, 13 candidate genes were obtained from the *L. chinense* genome, including five *LcGA20ox* and eight *LcGA2ox* genes. However, *GA3ox* was not detected in *L. chinense*. This result indicated that *LcGA3ox* was either lost during evolution or was the result of incomplete genome assembly. The *LcGA20ox 1–5* and *LcGA2ox 1–8* genes were named according to the order of their identification (Table 2). All of the genes were distributed on eight chromosomes except for *LcGA2ox2*, which was mounted to conting2730. The 1st, 6th, and 9th chromosomes had multiple genes, but only one gene was located on the 2nd, 3rd, and 9th chromosomes. The coding sequences (CDS) of *LcGA20ox* and *LcGA2ox* ranged from 732 bp (*LcGA2ox6*) to 1290 bp (*LcGA20ox2*). The length of the amino acid sequences ranged from 244 aa to 430 aa. The molecular weight of the *LcGAox* members ranged from 27.340 kDa to 48.364 kDa. The molecular weight differences of the *LcGA20ox* and *LcGA2ox* classes were 5.705 kDa and 20.348 kDa. The predicted values of the isoelectric points of *LcGAox* ranged from 5.31 to 8.60. The results suggest that *LcGA2ox1,2,6* and *LcGA20ox3* are basic proteins, whereas the remaining proteins are acidic. The predicted subcellular localization results showed that the *GA3ox* genes in *L. chinense* were expressed in the cytoplasm.

### 2.2. Evolutionary Analysis of LcGA20ox and LcGA2ox Genes

A phylogenetic tree was constructed based on the full-length protein sequences of *GA3ox* in *L. chinense* (Lc), *O. sativa* (Os), *A. thaliana* (At), *A. trichopoda* (Atr), and *V. vinifera* (Vv) (Figure 1). According to previous research [30], the sequences were divided into five subgroups, namely *GA20ox*, *GA3ox*, *GAox*, *C19GA2ox*, and *C20GA2ox*. As there were no *GA3ox* genes found in the *GA3ox* and *GAo*x subgroups of *L. chinense*, we will only refer to the phylogenetic relationships of *GA20ox*, *C19GA2ox*, and *C20GA2ox* in the five species. The results showed that *GA20ox*, *C19GA2ox*, and *C20GA2ox* in each species were distributed in different subfamilies (Figure 1). *LcGA20ox* included five members, namely *LcGA2ox1,2,3,6*, which belonged to *C19GA2ox*, and *LcGA2ox4,5,7,8* was classed into the *C20GA2ox* subfamily. The phylogenetic tree result indicated that most members of *LcGA2ox* and *LcGA20ox* were clustered into a clade with *VvGA20ox*, *VvGA2ox*, *AtrGA20ox*, and *AtrGA2ox*. This indicated that the *GA3ox* of *L. chinense*, *V. vinifera*, and *A. trichopoda* could have originated from the same ancestor, except for *LcGA2ox5*. In the *C19GA2ox* subgroup, *LcGA2ox1,2,3* shared high similarities, indicating that they could have functional consistency, and the other members in the different subfamilies may have formed functional specializations during evolution.

The evolution of gene function is related to changes in gene structure. The *GAox* protein conserved motif and gene structure of *L. chinense* and *A. thaliana* were analyzed (Figure 2 and Appendix A). The *GAox* family protein contained two conserved domains: 2OG-FeII oxygenase and DIOX_N, which play important roles in the synthesis of GA in plants [14]. To investigate the motifs shared among related proteins within the same subfamily, the 15 distinct motifs were predicted by MEME software (Figure 2 and Appendix A). In the prediction result, motifs 1, 2, 3, 5, 6, 8, 9, 11, and 12 were identified in most of the *GAox* proteins, except the individual *LcGAox*. Motifs 4, 13, and 14 only existed in *GA20ox*, and motif 7 was predicted in *GA20ox* and *GA3ox*. Motifs 10 and 15 were unique to *C19GA2ox* and *C20GA2ox*, respectively. Interestingly, motif 9 was located in the N-terminal of *GA20ox* and in the C-terminal of the other *GAox* proteins. Motifs 1, 7, and 12 belong to the DIOX_N domain, and motifs 2, 3, 5, 6, and 11 belong to the 2OG_FeII oxygenase domain. We found that some specific motifs were absent or increased in *L. chinense*. Motif 12 was absent in *LcGA2ox8*. Motifs 2 and 5 were absent in *LcGA2ox4*, and motif 11 was absent in *LcGA2ox6*. In addition to the motifs of the two domains, the specific motif 10 in *C19GA2ox* was identified in *LcGA20ox4*. The GA20 oxidase contained a conserved GA substrate-binding motif and 2-oxoglutarate-binding motif. According to previous research, the amino acid residues that bind the Fe^2+^ and interact with the 5-carboxylate of 2-oxoglutarate belong to GA oxidase [31,32]. The multiple sequence alignments and 3D structure results showed that the GA substrate-binding motif (NYYPPXCXXP) and 2-oxoglutarate-binding motif (LPWKET) are also ubiquitous in *LcGA20ox* genes (Figure 3). The amino acid residues with specific functions were conserved in *LcGAox* and *AtGAox*, except for *LcGA2ox4* and *6*, in which the HXD dyad near the amino terminus is lost (Figure 3 and Appendix A). Summarizing the above results, the *LcGAox* and *AtGAox* superfamily possesses common conserved domains and sites. However, the amino acid sequences of the domains are diverse, indicating that functional differentiation may be present between the two.

The gene structure analysis results of *GAox* in *L. chinense* and *A. thaliana* are indicated in Figure 3. The number of exons in most *GAox* genes was between 2 and 3. *AtGA3ox2* only had one exon, while *LcGA2ox5* and *LcGA20ox1* had four exons. Most *GAox* genes had similar gene structures, but were found to possess functional conservatism. Interestingly, *LcGA20x1* had a very long intron near the C-terminal (about 30 kb), which may be caused by genome annotation issues or the presence of transposable elements; an observation that needs to be further verified.

### 2.3. Synteny Analysis of GAox Genes in L. chinense, Grape, Arabidopsis, and Rice

Whole-genome duplications, segmental duplications, and tandem duplications play important roles in gene family expansion. Thus, we analyzed the duplication events of *GAox* in *L. chinense*. In the *L. chinense* genome, two pairs of segmentally duplicated genes were identified: *LcGA2ox3/LcGA2ox1* and *LcGA2ox3/LcGA2ox6* (Figure 4a). This result indicates that the *LcGA2ox* gene family was amplified via the segmental duplication of genes, whereas no duplication events were found in *LcGA20ox*. To further explore the evolutionary relationships of the *GAox* gene family, we performed collinearity analysis among *L. chinense* and three other species (*Arabidopsis*, grape, and rice). The results indicated that most of the orthologous *GAox* genes were distributed into multiple chromosomes, except in rice, which had only one location for Os-1 (Figure 4b). In Lc/At, Lc/Os, and Lc/Vv, the number of orthologous genes in Lc/Vv was much higher than in the other two groups, indicating that the relationship between grape and Liriodendron was closer in the *GAox* gene family.

To analyze the evolutionary selection in the *LcGAox* gene family, we counted the nonsynonymous (Ka) and synonymous substitutions (Ks) among the orthologous gene pairs, as well as the Ka/Ks ratios among the four species (Table 3). The results showed that the genes of the duplicate pairs had KA/Ks ratio < 1, with no ratio > 1, and the KA/KS ratio within species was less than the ratio between species. This suggested that the orthologous *GAox* genes of the four species were subjected to purifying selection during evolution.

### 2.4. Codon Usage Pattern Analysis in GAox Genes

The codon usage pattern reflects the evolution and mutation of species or genes and also plays an important role in gene expression level and functional differentiation [33,34]. By analyzing the *GA20ox*, *C19GA20x*, and *C20GA2ox* subgroup genes in 21 species (approximately 391 sequences; Appendix A), we constructed relatively systematic codon usage patterns in the gibberellin oxidase gene family. Codon usage bias (CUB) is largely affected by the whole base content of each gene, and the overall GC content often reflects the strength of directional mutation, and the main difference of the synonymous codon is typically reflected in the third base [35]. The basic statistics of the third base are indicated in Figure 5. The usage percentage of G/C in the third base was obviously less than A/T in the dicotyledons, and the G3s/C3s ratio was much higher than the A3s/T3s ratio in the monocotyledons. Among the magnoliaceous plants represented by *L. chinense* and the Amborella, the usage percentage of G3s/C3s and A3s/T3s was almost equal. The third base usage model among mosses and lycophytes was similar to the monocots. This suggested an unequal distribution of A, T, C, and G, as well as different biases during plant evolution. We estimated Karl Pearson’s correlation coefficients between the nucleotide composition at the third codon position and GC12 to analyze the codon usage pattern in the *GAox* gene family (Figure 5). In all 21 species, significant positive correlations were detected between C3s, G3s, GC, GC3s, and GC12s, indicating that the codon usage pattern was selected by mutation pressure in the *GAox* gene family. The ENC is an effective indicator for evaluating the overall codon preference of codon genes, and we used ENC as the ordinate and GC3s as the abscissa to construct an ENC plot to explore the relationship between base composition and codon preference (Figure 6a). The result showed that the ENC values of *GA20ox* and *GA2ox* ranged from 40 to 60, except for in the monocotyledons. Most of the *GAox* genes in the mosses, lycophytes, Amborellaceae, and *L. chinense* (Magnoliaceae) had ENC values that were basically around 50–60, showing no obvious codon bias (Figure 6a). However, for *GA20ox* and *GA2ox* in the monocots, the gene ENC values were generally distributed around 35, except for some *C19GA2ox* genes, indicating that the gene codons had significant usage preference. ENC plots are widely used to analyze the main factors affecting codon usage [28]. Most of the points were in close proximity to the curve (Appendix A), indicating that mutation pressure played an important role in the codon usage bias, though natural selection and other factors could have had a certain influence. Parity rule 2 (PR2) is considered to be an important index that assesses whether base bias in different mutation and selection pressures exists in the nucleotide composition. In the *GA20ox* and *C19GA2ox* gene subgroups, the third position of the codon T was used more frequently than A, and C was used more frequently than G (Figure 6b). The frequency of nucleotide A of *LcGA20ox* was higher than that of T, but this result was in contrast to that of the *LcC19GA2ox* genes. The trend of base usage frequency of *L. chinense* was consistent with the overall *C20GA2ox* gene subfamily and was in the order of T > A, G > C. In summary, A and T were not equal to G and C, and a few genes were close to the center, suggesting that natural selection and mutation pressure might have influenced the CUB of the *GA20ox*, *C19GA20x*, and *C20GA2ox* gene subfamily.

We performed RSCU analysis of codons to better understand the patterns of codon usage in the *GAo*x gene family. Among the 59 synonymous codons in the *GA20ox* and *GA2ox* gene families of all 21 species (Appendix A), we found that some codons exhibited no obvious preference (RSCU = 1), but in different subgroups, some special codons were found in different species. For example, AAG, CAC, GAT, and GAA had significant preferences (RSCU > 2) in Cruciferae. In the *GA2ox* subgroups, AGG and AGA (RSCU > 2), exhibited usage bias in most species, except for monocots (Figure 7).

The above analysis results indicated that the *GA20ox*, *C19GA2ox*, and *C20GA2ox* gene subgroups exhibited comparatively more bias towards A/T-ending codons compared to G/C-ending codons in most species, but the monocotyledon nucleotide composition biases of the third were conversed with the other species, with strong codon preference. This base composition pattern, which differs significantly from dicotyledonous plants, may be due to mutation pressure and natural selection. *Liriodendron chinense* had similar base composition patterns as *A. trichopoda*, as well as no obvious preference for codon usage an aspect that differs between monocotyledonous and dicotyledonous plants, thus indicating that the *LcGAox* genes may be more primitive.

### 2.5. Cis-Elements in the Promoters of LcGA2ox and LcGA20ox

The cis-regulatory element in the promoter region determines the binding specificity of the transcription factor, and, thus, it has the role of transcription regulation. PlantCARE was used to predict the cis-elements related to stress responsiveness and phytohormone responsiveness to analyze the functions of *LcGA20ox* and *LcGA2ox* (Figure 8). We found that the G-Box and ABRE motifs were ubiquitous and relatively abundant in all the gibberellin oxidases of *L. chinense*, and in comparison, with other *LcGAox* genes, the number was higher in the promoters of *LcGA20ox2*, *LcGA20ox5*, *LcGA2ox1-4*, and *LcGA2ox7* (Figure 8a). This indicated that the ABRE and G-Box motifs have major functions in the response to stressful environments. These cis-elements were mainly involved in ABA, GA, SA, and auxin responsiveness, as well as light, low temperature, and other stresses (Figure 8b). In general, light-responsive elements are widely present in the *LcGAox* gene family, whereas abiotic stress elements are comparatively fewer and only exist in *LcGA2ox2*, *4,5,8, LcGA20ox2,4,* and *5*. Different hormones also played a regulatory role in the *LcGAox* genes. *LcGA2ox1,2,4,5,8, LcGA20ox2,4,* and *5* were regulated by SA, and eight genes, including *LcGA2ox2,3,4,5,7*, *LcGA20x1,2,* and *3*, were related to auxin. Most genes, except for *LcGA2ox8*, were associated with ABA. Although all genes are related to gibberellin oxidase, some genes did not contain gibberellin responsiveness elements in their promoters, including *LcGA2ox5* and *LcGA20ox3.*

### 2.6. Analysis of the Expression Patterns of LcGA2ox and LcGA20ox Genes in Different Organs

The transcriptome sequencing data of *L. chinense* were downloaded from the NCBI SRA database, and the gene expression level was obtained using salmon analyses. To elucidate the expression patterns of the *LcGA20ox* and *LcGA2ox* genes in *L. chinense* growth and development, we constructed a gene expression profile in different stages of hybrid *Liriodendron* somatic embryogenesis and different organs of *L. chinense* (Figure 9). *LcGA2ox1,2,8* and *LcGA20ox3* were involved in the process from embryogenic callus to seedling morphogenesis, and *LcGA2ox8* exhibited a trend of high expression levels. During the transition from globular embryo to cotyledon embryo, the expression level of *LcGA20ox3* was downregulated (Figure 9a). *LcGA20ox1,4,5, Lc*GA2ox*3,5,* and *6* exhibited no or very low expression in the nine organs (Figure 9c). High expression levels of *LcGA2ox2* were detected in LP7 (unexpanded leaf) (Figure 9). The expression levels of *LcGA2ox7* and *LcGA20ox2* were obviously upregulated in the pistils and stamens in comparison with the other organs. *LcGA20ox1* was only detected in LP2 and LP7, while *LcGA20ox3* and *LcGA2ox4* and *8* possessed higher expression levels in the mature leaves, indicating that the different types of gibberellin oxidases participated in the development of the leaves. In the bracts, sepals, petals, and stamens, *LcGA20ox3* and *LcGA2ox4* and *8* were expressed to varying degrees to regulate the balance of internal gibberellin. *LcGA20ox3* and *LcGA2ox4* and *8* were also detected at the different developmental stages of the petals. The *LcGA2ox8* and *LcGA20ox3* expression levels remained stable, while the expression level of *LcGA2ox4* was inconsistent with *LcGA2ox8*, exhibiting low expression after the S1 stage (Figure 9b). These results showed that the synthesis and decomposition of gibberellin were regulated by different gibberellin oxidases in the vegetative and reproductive organs of *L. chinense*.

### 2.7. Effects of Phytohormone and Abiotic Stress Influence the Expression of LcGA2ox and LcGA20ox

To investigate whether the *LcGA20ox* and *LcGA2ox* gene families participate in the abiotic stress and phytohormone response, the gene expression levels in the different organs of three-month-old seedling were measured under NaCl, PEG, 4 °C, GA, ABA, and CCC (chlormequat chloride) treatments by qRT-PCR. The results showed that only *LcGA2ox1*, *LcGA2ox4*, and *LcGA2ox7* were involved in the above stresses and exhibited a similar expression pattern under most of the treatments (Figure 10).

Under abiotic stress, the expression levels of *LcGA2ox1* showed an upward trend at first followed by a downward trend under the same treatment (Figure 10 and Appendix A). The expression levels of *LcGA2ox1* and *LcGA2ox4* reached their peaks at 6–24 h in the roots, stems, and leaves under PEG and low temperature stress. Under NaCl stress, the *LcGA2ox1* expression level in the roots and leaves peaked at 12 h. Compared with untreated plants, *LcGA2ox1* responded to different abiotic stresses far more than normal level (Log2FD > 3) (Appendix A). The *LcGA2ox4* expression level in the leaves was downregulated compared with *LcGA2xo1* and was hardly detectable in the roots and stems. After 24h of NaCl stress treatment, *LcGA2ox4* expression showed a certain degree of response (Figure 10 and Appendix A). It was worth noting that under low temperature stress, *LcAG2ox4* had a relatively specific response in leaves, and it has always maintained a relatively stable state. Although *LcGA2ox7* had different expression patterns under different stresses, the gene relative expression levels changed little overall. 

Under the GA and ABA treatments (Figure 10 and Appendix A), the expression level of *LcGA2ox1* and *LcGA2ox4* in the leaves peaked at 24 h or 12 h, respectively. *LcGA2ox7* was detected after 48 h and 6 h of ABA and GA treatment in the leaves. Almost no expression of *LcGA2ox1/4* can be detected in the stem. However, the *LcGA2ox7* expression pattern in the stems was interesting, as within 12 h after ABA and GA treatment, the expression level was downregulated compared with at 0 h in the stems. However, the gene expression level suddenly increased at 24 h, following which it began to decline. 

After CCC treatment (Figure 10 and Appendix A), *LcGA2ox4* and *LcGA2ox7* expression was suppressed in the stems, but the expression level of *LcGA2ox1* was upregulated in the stems, with the expression reaching the highest level at 6 h, after which it was downregulated. It was indicated that the active gibberellin or precursor substance degraded by *LcGA2ox1* may be different from *LcGA2ox4/7*. In the leaves, the response of *LcGA2ox1* and *LcGA2ox4* to CCC was relatively great, peaking at 12 h, following which the expression level decreased.

The above results indicated that in different organ parts of hybrid *Liriodendron*, the *LcGA2ox1/4/7* can respond to abiotic stress and hormone treatment to varying degrees, thereby regulating the balance of gibberellin in the plant to maintain the normal growth.

## 3. Discussion

GA is an important hormone that controls plant growth and development [1,2,3,4], thus, its related metabolic and synthetic pathways are of great research interest. As an important member of the 2–oxoglutarate-dependent dioxygenase superfamily, gibberellin oxidase catalyzes the last step of gibberellin metabolism, and its related functions have been verified in many species [7,16,24,26,28].

### 3.1. Gene Identification, Phylogenetic Relationship, Motif Analysis, Gene Structure, and Collinearity Analysis

In this study, we identified 13 gibberellin oxidase genes belonging to the *GA20ox*, *C19GA2ox*, and *C20GA2ox* subfamilies. However, we did not detect *GA3ox* in the *L. chinense* genome. This shows that *LcGA3ox* may have been lost during the evolutionary process. A phylogenetic tree was constructed based on the 72 *GAox* protein sequences belonging to four subgroups, and the 13 members clustered into three subgroups. Interestingly, *AtrGA20ox4* and *OsGAox5, 7,* and *8* clustered together, indicating that *AtrGA20ox4* may occupy a special position in evolution based on previous research [36]. The protein sequences of *LcGA2ox1,2,* and *3* had high homology, suggesting that functional redundancy may exist between the three of them. Most *LcGAox* members clustered together with or close to the members of *VvGAox* and *AtrGAox*, implying that they are closely related and may have originated from a common ancestor. Mutations of gene function are often related to changes in gene structure and conserved domains [37]. By comparing the gene structure and conserved functional domains between *Arabidopsis* and *L. chinense*, we obtained information about the similarities and differences between the two *GAox* gene families. *LcGAox* and *AtGAox* had high similarity in gene structure, indicating that there was a certain degree of conservation between the two families, but the results of the motif analysis showed that some *LcGAox* (Figure 2) genes had lost or added the domain of 2OG-FeII oxygenase and DIOX_N [27], which may lead to gene functional changes. For example, motifs 12, 2, 5, and 11 were absent in *LcGA2ox4, 6*, and *8*, and the specific motif 10 in *C19GA2ox* was found in *LcGA20ox4*. Gene duplication events, as the main factor of genome expansion, mainly include tandem duplication and fragmented duplication [38]. Two fragmented duplications were identified in the *LcC19GA2ox* gene family, showing that this family experienced an expansion, which may be related to a single whole-genome duplication event in *L. chinense*. Collinearity analysis with members of the *GAox* genomic group in grapes, rice, and Arabidopsis showed that there were multiple orthologous genes with L. chinense, and the number of orthologous genes with grape was the largest, indicating a close relationship between the *LcGAox* and *VvGAox* gene family. The results of the evolutionary selection analysis indicated that the *GAox* genes in the five species experienced purifying selection.

### 3.2. Codon Usage Pattern and cis-Element Analysis

Through analysis of the *GAox* codon usage patterns of 21 species, we found that the *GAox* of angiosperms had obvious codon preference. Among them, the third base of the amino acid encoded by monocotyledons was biased towards A/T, while that of dicotyledonous plants was biased towards C/G, and this bias was affected by many factors, such as natural selection and mutation pressure. However, *LcGAox* exhibited no such strong bias. This situation was similar to that of *Amborella* and *Lycopodium*, which also showed that *LcGAox* was more evolutionarily primitive [39]. By analyzing the cis-elements related to stress and phytohormones in the *LcGAox* promoter, we found that light-responsive elements were widely present in the promoter region. *LcGA2ox1-4, 6*, and *7* and *LcGA20ox1* and *3* do not directly respond to external abiotic stress, and no GA responsive element was predicted in *LcGA2ox5* and *LcGA20ox1* and *3*, indicating that they are related through other factors.

### 3.3. The Spatial Expression Pattern of LcGAox and Their Response to Abiotic and Phytohormone Stress Base on the RNA-seq and qRT-PCR

Gibberellin mainly acts on young organs, such as the seeds, young leaves, and apical buds, et al. [11,14]. The transcriptome data of multiple organs indicated that *LcGA20ox3*, *LcGA2ox8*, and *LcGA2ox4* were ubiquitously stably expressed in the different organs. *LcGA2ox1,2* exhibited stable expression in the developmental process from globular embryo to cotyledon embryo and young leaves. The above results indicated that plants complete the synthesis and degradation of gibberellin by regulating the two genes *GA20ox* and *GA2ox*, so as to achieve an optimal steady state at a specific organ stage. Interestingly, we found the only *LcAG20ox3* played an important role in the *LcGA20ox* gene family, showing that the other *LcGA20ox* members either had a specific response or just played an auxiliary role in the synthesis of GA. Combined with the prediction results of the homeostatic regulatory elements and previous studies, it is indicated that gibberellin oxidase has different response modes in response to some stress conditions and hormone treatments. *LcGA2ox1* and *4* responded rapidly to PEG and low temperature stress in the roots, stems, and leaves. Under salt stress, the expression of *LcGA2ox1* in the roots, stems, and leaves peaked within 12 h, while *LcGA2ox4* exhibited a lower response in the leaves. Under the stimulation of ABA, *LcGA1*, *4*, and *7* had varying responses in the leaves, and *LcGA2ox4* specifically responded to ABA in the stems and upregulated the expression level. The application of exogenous GA activated the expression of *LcGA2ox1, 4, 7* in the leaves and *LcGA2ox7* in the stems, thereby degrading GAs. Under the treatment of a GA inhibitor, the expression of *LcGA2ox1,4* was upregulated in the leaves compared with at 0 h, while the expression of *LcGA2ox7* was downregulated in the stems. These results indicated that *LcGA2ox1* and *4* mainly degraded the GA in the leaves, while the degradation of GA in the stems mainly depended on *LcGAox7*, which was indicative of functional specificity.

In summary, our research explored the functional differences of the *GAox* family in different organs of *L. chinense*, and our results indicate their potential role in the evolution of *GAox* genes, an aspect worth further study.

## 4. Materials and Methods

### 4.1. Identification of Gibberellin Oxidase GA2ox, GA20ox Genes in the L. chinense Genome 

To identify the *LcGA2ox, LcGA3ox*, and *LcGA20ox* gene families in *L. chinense*, the *Arabidopsis*
*GA2ox*, *GA3ox*, and *GA20ox* protein sequences were downloaded from phytozome v12.1 (https://phytozome.jgi.doe.gov/pz/portal.html#, accessed on 12 June 2021), and the conserved OG-FeII_Oxy (PF03171) and DIOX_N (PF14226) were used as a query. BLASTP and HMMER3.0 software were used to search the target protein sequences in the L. chinense protein database (https://hardwoodgenomics.org/Genome-assembly/2630420, accessed on 12 June 2021). The LcGA20x, Lc*GA2ox*, and Lc*GA3ox* protein sequences were further authenticated based on the conserved domains using SMART (http://smart.emblheidelberg.de, accessed on 12 June 2021) and CDD-search (https://www.ncbi.nlm.nih.gov/Structure/bwrpsb/bwrpsb.cgi, accessed on 12 June 2021). After removing redundant proteins, a total of 213 putative proteins were identified. A phylogenetic tree was constructed using the 213 identified proteins from *L. chinense* and 16 *GAox* proteins from *Arabidopsis*. A total of 13 *LcGAox* proteins were obtained (Appendix A). The gene properties, including the molecular weight (kDa) and isoelectric point (pI) of each protein, were determined using the Compute pI/Mw tool on the ExPASy website (Table 1). The subcellular localization of the *LcGAox* genes was predicted by Cell-PLoc 2.0 (http://www.csbio.sjtu.edu.cn/bioinf/Cell-PLoc-2/, accessed on 12 June 2021).

### 4.2. Phylogenetic Tree Construction and the Analysis of Gene Structure, Conserve Motif in Multiple Species and the 3D Structure Determination

The *GAox* protein sequences were download from phytozome in Oryza sativa (Os), Amborella trichopoda (Atr), and Vitis vinifera (Vv) based on the *AtGAox* protein sequences (Table 1 and Appendix A). The phylogenetic relationships between *LcGAox* and the *GAox* proteins from the other four species were determined using the IQtree software (http://www.iqtree.org/, accessed on 12 June 2021). Sequence relationships were inferred using the maximum likelihood (ML) method, and the optimal model was LG + F + R6 (Figure 1). The *LcGAox* sequences were named according to their respective clades. The gene structure information of *AtGAox* was obtained from the *Arabidopsis* genomic annotation database in phytozome. The conserved motifs of the *LcGAox* and *AtGAox* proteins were checked using the online multiple expectation maximization for motif elicitation (MEME) program. A hierarchical clustering tree was constructed using the neighbor-joining algorithm with the default parameters of 1000 bootstrap replications in the MEGA 7.0 software [40]. Images were illustrated using the TBtools software [41]. The three-dimensional (3D) structure was predicted using the online tool SWISS-MODEL (https://swissmodel.expasy.org/interactive, accessed on 12 June 2021).

Chromosomal distribution and gene duplication most of the *LcGA2ox* and *LcGA20ox* genes were mapped to the *L. chinense* chromosomes based on the location information from the genome database of *L. chinense* by TBtools. The gene repetitive events were analyzed by the Multicollinearity ScanToolkit (MCScanX) [42]. The synteny relationship map of the orthologous *GAox* genes obtained from *L. chinense*, *Arabidopsis*, grape, and rice (Kitaake) were visualized by TBtools. The evolutionary selection relationships of orthologous genes within and between species were analyzed in DnaSP6.

### 4.3. Codon Usage Pattern Analysis in GAox Genes

To systematically analyze the codon usage pattern of *GAox* genes, we used *AtGAox* genes as a query in the phytozome database to identify all *GAox* genes from *Physcomitrella patens*, *Selaginella moellendorffii*, *Amborella trichopoda*, *Sorghum bicolor*, *Oryza sativa*, *Zea mays*, *Setaria italica*, *Arabidopsis thaliana*, *Brassica oleracea*, *Zea mays*, *Medicago truncatula*, *Gossypium Raimondii*, *Theobroma cacao*, *Manihot esculenta*, *Ricinus communis*, *Citrus sinensis*, *Malus domestica*, *Populus trichocarpa*, *Cucumis sativus*, *Solanum lycopersicum*, and *Vitis vinifera* using BLAST (Appendix A). The ML method was used to identify different group members by constructing evolutionary trees. The coding sequences of the *LcGA2ox* and *LcGA20ox* genes were used to calculate the A, G, C, T, and GC contents at the third site of the synonymous codon (A3s, G3S, C3s, T3s, GC3s content), relative synonymous codon usage (RSCU), and effective numbers of codons (ENC) with the CodonW v.1.4.2 (Appendix A). Correlation analysis between codon composition and preference parameters (A3s, T3s, G3s, C3s, GC3s, GC) was carried out using R (www.r-project.org, accessed on 12 June 2021).

### 4.4. Analysis of the cis-Acting Element in LcGA2ox and LcGA20ox Gene Families

The promoter sequences of *LcGA2ox* and *LcGA20ox* were extracted from the *L. chinense* genome database (3000 bp), and the cis-acting elements were predicted and analyzed using the PlantCARE online site [43]. 

### 4.5. Analysis of Gene Expression in Organs by the RNA-seq

To analyze the expression patterns of the *LcGA2ox* and *LcGA20ox* gene families, different organ transcript data for *L. chinense* were downloaded from the NCBI Sequence Read Archive (SRA) with the accession numbers SRR8101040, SRR8101041, SRR8101042, SRR8101043, SRR9945429, SRR9945430, SRR9945433, SRR9948913, SRR9948914, SRR9948915, SRR9948916, SRR9948917, SRR9948918, SRR9948919, SRR9949005, SRR9949006, SRR9949007, SRR9949008, SRR9949009, and SRR9949010. All of the mRNA abundance values were measured by transcripts per million (TPM) based on the *L. chinense* genomic database [28] and the transcript data (TPM) of *LcGA2ox* and *LcGA20ox* were indicated in a heatmap.

### 4.6. Plant Materials Treatment and Expression Analysis by qRT-PCR

Three-month-old somatic embryo seedlings were cultured in an incubator under white light (16 h light, 8 h dark) on 3/4MS medium. They were treated separately with 100 mg/L abscisic acid (ABA), 100 mg L^−1^ GA, 100 mg L^−1^ chlormequat (CCC), 20% polyethylene-glycol (PEG)-6000 solution, 100 mM NaCl solution, or low temperature (4 °C). Mature leaves, stems, and roots were sampled from three biological replicates of the treatment and control plants at 0, 6, 12, 24, and 48 h after the trial. Quantitative RT-PCR analysis was used to confirm the expression patterns of *LcGA2ox* and *LcGA20ox* in the different organs under the different treatments (Appendix A). Total RNA extraction was performed using a KK Fast Plant Total RNA Kit. First-strand cDNA was synthesized from 1.0 mg of RNA with an Evo M-MLV RT Kit with Gdna Clean for qPCRII AG11711 (Accurate Biotechnology (Hunan) Co., Ltd.). The qRT-PCR was carried out using SYBR-green fluorescence in a Roche LightCycler^®^480 Real-Time PCR System. The ΔΔCT method was used to calculate the gene relative expression levels [44]. All qRT-PCR primers were designed by Primer5.0 and are listed in Appendix A.

### 4.7. Data Analysis

All data analysis was based on Excel 2019, analysis of variance was based on R, and multiple comparison method was least significant difference (LSD)

## 5. Conclusions

In this study, we identified 13 gibberellin oxidase genes based on publicly-available *L. chinense* genomic data, which included five *GA20ox* genes, four *C19GA2ox* genes, and four *C20GA2ox* genes. Phylogenetic analysis, gene structure, and conserved motif, and functional site prediction analyses revealed that gibberellin oxidases in *L. chinense* are conserved and divergent compared with other species. The codon usage pattern indicated that the *LcGAox* genes had no obvious codon usage bias and were more biased towards primitive species compared with angiosperms. The RNA-seq and qRT-PCR data analysis showed that *LcGAox* genes exhibit specific expression patterns in different organs and the stages of somatic embryogenesis. Furthermore, *LcGA2ox1, 4,* and *7* can further affect the synthesis or degradation of gibberellin in hybrid *Liriodendron* by responding to abiotic stress or hormone treatment. In summary, this study provides a basis for further investigating the genetic and functional characteristics of the *LcGAox* gene family.

## Figures and Tables

**Figure 1 ijms-22-07167-f001:**
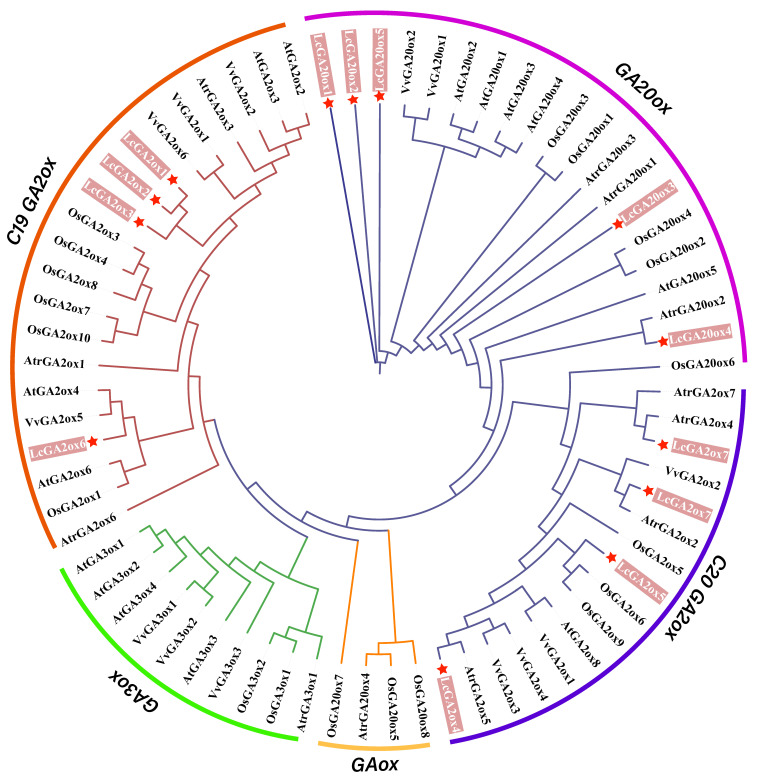
Phylogenetic analysis of gibberellin oxidase in *L. chinense* (Lc), *O. sativa* (Os), *A. thaliana* (At), *A. trichopoda* (Atr), and *V. vinifera* (Vv). The amino acid sequences of *GA20ox*, *GA2ox*, and *GA3ox* were aligned with ClustalX, and phylogenetic tree was constructed using maximum likelihood (ML) method in IQtree2.0.5 with 1000 bootstrap replicates, the LG + F + R6 was selected to the optimal model. The gibberellin oxidase genes in *L. chinense* were showed by star in red.

**Figure 2 ijms-22-07167-f002:**
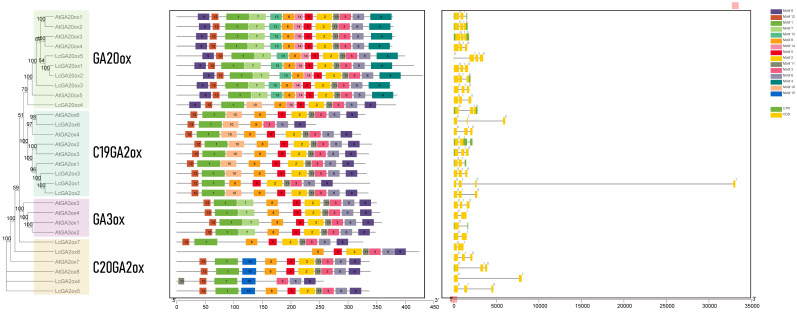
Evolutionary relationships, conserved protein motifs and gene structure in *GAox* genes from *L. chinense* and *A. thaliana*. The phylogenetic tree was constructed using the neighbor joining (NJ) method with 1000 bootstrap replicates in MEGA 7.0. All protein sequences of gibberellin oxidase in *L. chinense* and A. thaliana were divided to four subgroups. Different colors represent different subgroup. The numbers 1–15 of motif composition was displayed in different colored boxes in *LcGAox* and *AtGAox* proteins. Gene structure was showed off in the right box, the yellow box represents the coding sequences and the green box represent the UTR sequences.

**Figure 3 ijms-22-07167-f003:**
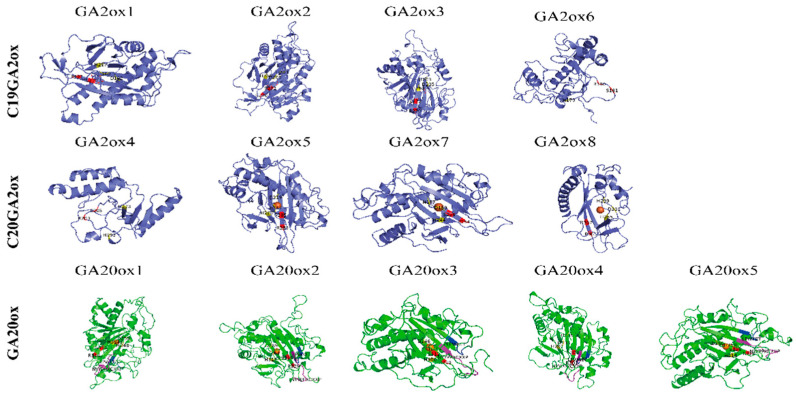
3D structures of LcGA oxidases showing functional sites. The conserved motif related GA substrate binding site and 2-oxoglutarate-binding motif were display the purple and blue bands. The amino acid residues that bind the Fe^2+^ and interacted with the 5-carboxylate of 2-oxoglutarate are highlighted in yellow and red, respectively.

**Figure 4 ijms-22-07167-f004:**
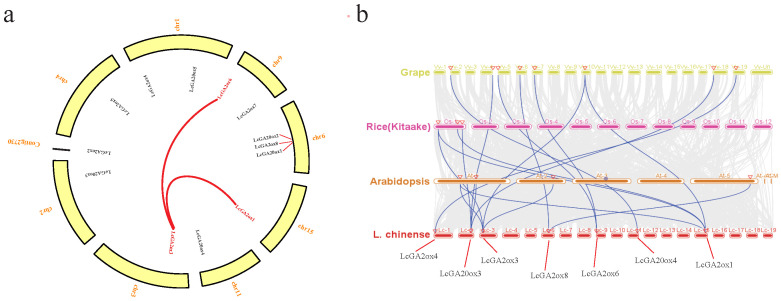
Genome-wide synteny analysis of *GA2ox* and *GA2ox* gene family among *L. chinense* and other three species. (**a**): Distribution and segmental duplication of *GAox* genes in *L. chinense*. The yellow panel shows the 8 chromosomes using a circle, and the contig are display through the black line. Red lines connecting homologous genes; chromosome numbers are marked outside of the circle. (**b**): Synteny analysis of *GAox* genes between *L. chinense*, *Arabidopsis*, grape, and rice (Kitaake). Gray lines in the background indicate the collinear blocks within *L. chinense*, *Arabidopsis*, grape, and rice (Kitaake) genomes, while the blue lines highlight the syntenic *GAox* gene pairs. Different gene pairs were highlighted by red triangle.

**Figure 5 ijms-22-07167-f005:**
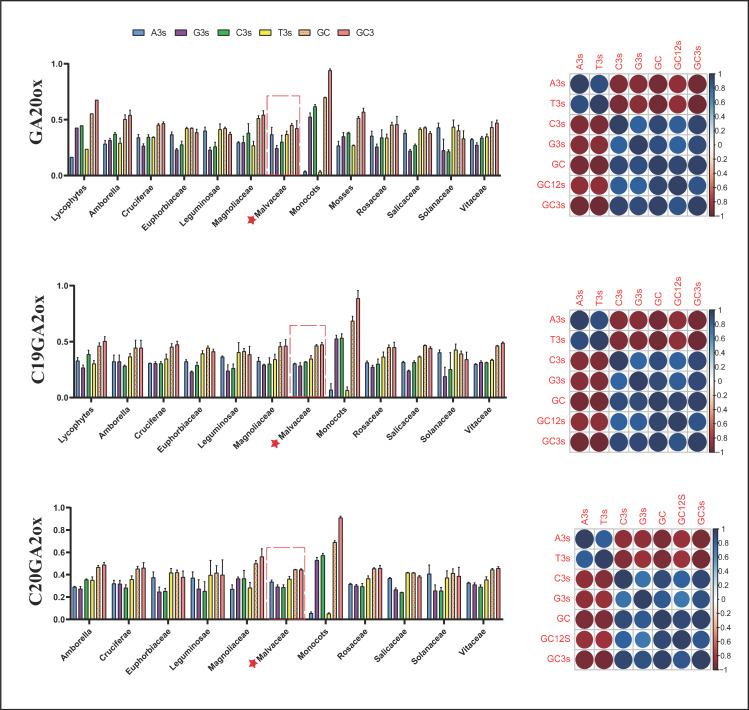
The bases in the third statistics and correlation analysis of *GA20ox*, *C19GA2ox*, and *C20GA2ox* in 21 species. The different color represents the third base frequency in codon. The red star and red dotted frame represent the L. chinense frequency of related bases in *GA20ox*, *C19GA2ox*, and *C20GA2ox*.

**Figure 6 ijms-22-07167-f006:**
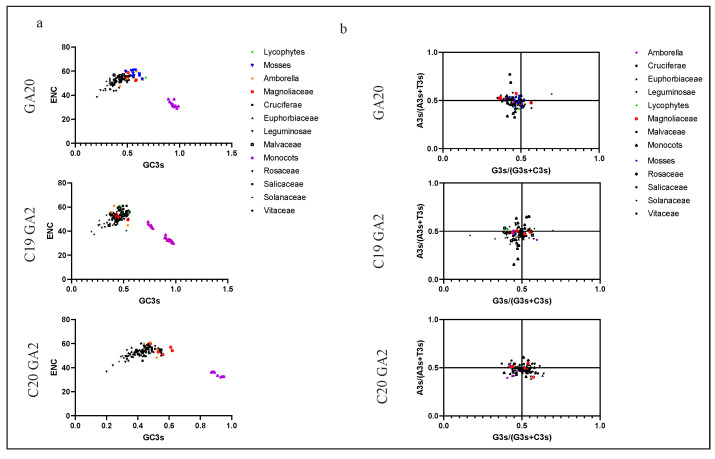
ENC-plot analysis and PR2-plot analysis in *GA20ox*, *C19GA2ox*, and *C20GA2ox*. (**a**): The points represent a single gene, ENC value of each gene as the ordinate and the GC3s value as the abscissa. Different shapes represent different families, and blue inverted triangle represents mosses, the green diamond represents lycophytes, the organ filled circle represents Amborella, the purple triangles represent monocots and the hollow red circle represents *L. chinense*. (**b**): This plot is tested with AT bias (3s/A3s + T3s) in the y-axis and GC bias (G3s/(G3s + C3s) in the x-axis in a graphical presentation. Different color and shapes represent different families.

**Figure 7 ijms-22-07167-f007:**
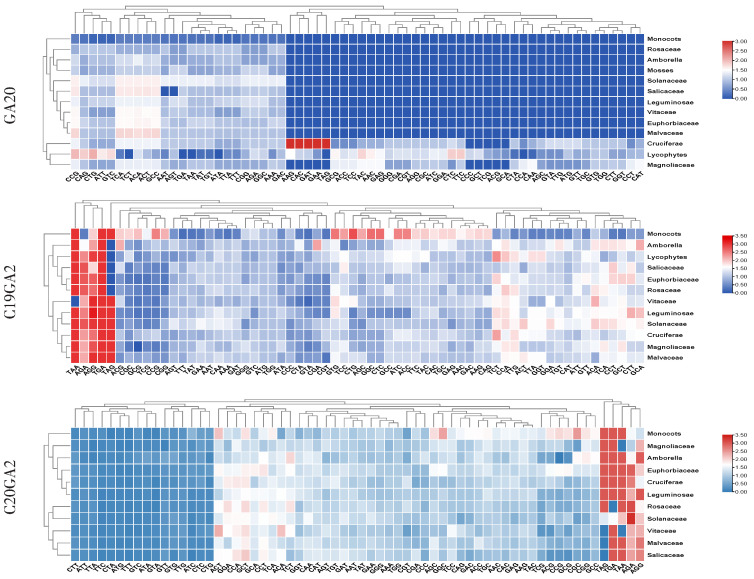
Relative synonymous codon usage (RSCU) of *GA20ox*, *C19GA2ox*, and *C20GA2ox* gene families in all 21 species. Blue-to-red color indicates low to high RSCU values of codons.

**Figure 8 ijms-22-07167-f008:**
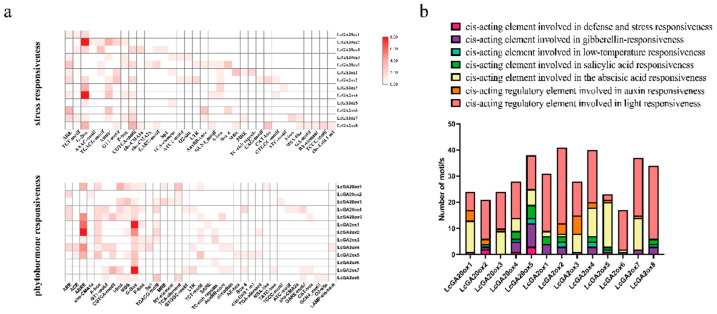
The number of cis-elements in *LcGA20ox* and *LcGA2ox* promoter. (**a**): The combine motifs about phyhormone and stress responsiveness in *LcGA20ox* and *LcGA2ox* gene families. The shade of the red represents the quantity. (**b**): The number of cis-elements about different hormone and abiotic stress in *LcGA20ox* and *LcGA2ox* gene families. Purple represents the gibberellin response element; Pink represents the light response element; Organ represents the auxin response element; Creamy yellow represents the abscisic acid response element; Green represents the salicylic acid response element; Light cyan represents the low-temperature response element; Purple represents the defense and stress response element.

**Figure 9 ijms-22-07167-f009:**
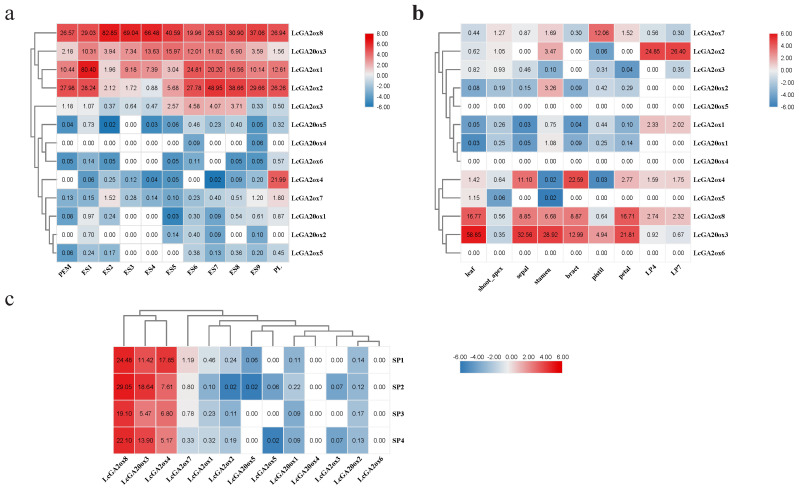
The *LcGA20ox* and *LcGA2ox* gene expression profiles in different organs. Two or three biological repeats were shown in the heatmap except bract. The transcripts per million (TPM) was used to indicated the gene expression level. PEM: embryogenic callus; ES1: after 10 days of liquid culture; ES2: two days after sieving; ES3: one day after ABA treatment; ES4: three days after ABA treatment; ES5: globular embryo; ES6: heart shaped embryo; ES7: torpedo embryo; ES8: immature cotyledon embryo; ES9: mature cotyledon embryo; PL: planta; SP1-4: the stage of petals development. (**a**) and (**c**) respectively represent the *LcGAox* gene expression level during somatic embryogenesis of hybridization liriodendron and the development stage of petal. These RNNA-seq data belong to the lab not publish the data. (**b**) represents the *LcGAox* gene expression level in different organs, and the accession numbers has listed in the material and methods 4.5. The expression levels of the genes were expressed by Transcripts Per Million (TPM).

**Figure 10 ijms-22-07167-f010:**
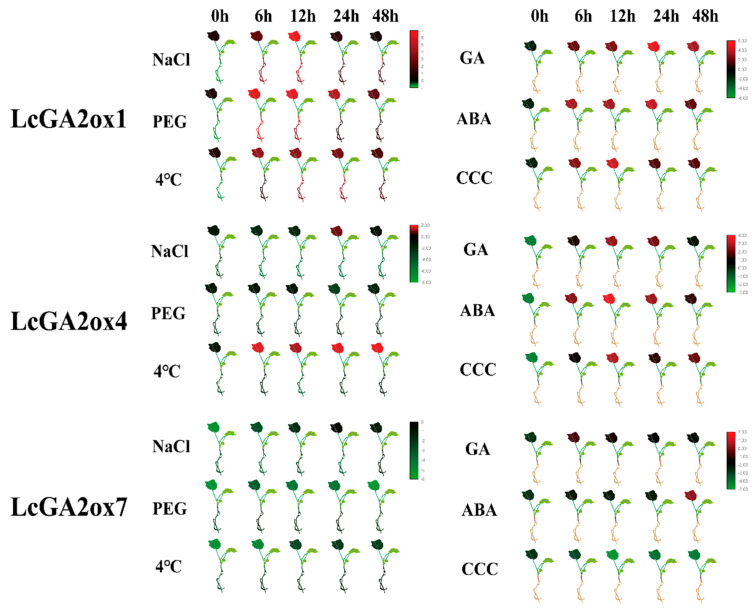
The gene expression level of *LcGA2ox1,4,* and *7* in leaves, stems, and roots under NaCl, PEG, 4 °C, ABA, GA, and CCC stress by qRT-PCR. The mean expression value was calculated from three independent biological replicates in different samples. The gene expression level was visualized by TBtools; The raw data of relative expression values and standard errors is provided in Appendix A.

**Table 1 ijms-22-07167-t001:** Number of *GAox* genes in five species.

Species	*GA20ox*	*GA2ox*	*GA3ox*	*GAox*	Total
*O. sativa*	5	11	2	3	21
*A. thaliana*	5	7	4	0	16
*A. trichopoda*	4	5	1	1	11
*V. vinifera*	2	6	3	0	11
*L. chinense*	5	8	0	0	13

**Table 2 ijms-22-07167-t002:** Summary of *LcGA0x* gene family characters.

Gene ID	Gene Name	Accession Number	Position	Location	CDS (bp)	Peptide (aa)	Mw (kDa)	pI	Subcellular Localization
Lchi18432	*LcGA20ox1*	MZ361712	Chr6	41600656-41604092	1245	415	46.976	5.93	Cytoplasm
Lchi25954	*LcGA20ox2*	MZ361709	Chr6	38707502-38709176	1290	430	48.367	5.97	Cytoplasm
Lchi01764	*LcGA20ox3*	MZ361716	Chr2	73946076-73948031	1125	375	42.662	8.36	Cytoplasm
Lchi01394	*LcGA20ox4*	MZ361713	Chr11	56684039-56686083	1149	383	43.514	5.78	Cytoplasm
Lchi10458	*LcGA20ox5*	MZ361707	Chr1	79200174-79213961	1197	399	45.202	6.1	Cytoplasm
Lchi04083	*LcGA2ox1*	MZ361710	Chr15	63254044-63287148	1014	338	38.043	8.18	Cytoplasm
Lchi35277	*LcGA2ox2*	MZ361718	Contig2730	15814-18566	1005	335	37.624	7.59	Cytoplasm
Lchi07023	*LcGA2ox3*	MZ361714	Chr3	21930812-21932403	996	332	37.241	6.41	Cytoplasm
Lchi12922	*LcGA2ox4*	MZ361717	Chr1	8453778-8461759	771	257	29.734	6.34	Cytoplasm
Lchi29789	*LcGA2ox5*	MZ361719	Chr4	67227159-97231797	1011	337	38.25	5.83	Cytoplasm
Lchi00133	*LcGA2ox6*	MZ361715	Chr9	2749738-2755769	732	244	27.34	8.6	Cytoplasm
Lchi15175	*LcGA2ox7*	MZ361708	Chr9	52008101-52009229	978	326	36.906	5.76	Cytoplasm
Lchi18410	*LcGA2ox8*	MZ361711	Chr6	42633756-42642514	1272	424	47.688	5.31	Cytoplasm

**Table 3 ijms-22-07167-t003:** The orthologous gene pairs’ Ka/Ks ratios among four species.

Duplicated *GAox* Gene Pairs			Ka	Ks	Ka/Ks
Lc/Lc	*LcGA2ox1*	*LcGA2ox3*	0.1752	0.7673	0.2283
	*LcGA2ox3*	*LcGA2ox6*	0.5347	1.5333	0.3487
Lc/At	*LcGA2ox1*	AT1G30040 (*AtGA2ox2*)	0.7197	0.9484	0.7589
	*LcGA20ox3*	AT1G44090 (*AtGA20ox5*)	0.5312	1.0585	0.5018
	*LcGA2ox8*	AT5G58660	0.6419	0.8638	0.7431
Lc/Vv	*LcGA2ox4*	VIT_219s0177g00020	1.4737	1.5654	0.9414
	*LcGA20ox4*	VIT_202s0234g00010	0.2196	0.686	0.3201
	*LcGA2ox1*	VIT_210s0003g03490	0.5232	0.6875	0.761
	*LcGA20ox3*	VIT_218s0001g01390	0.2933	0.525	0.5587
	*LcGA20ox3*	VIT_204s0044g01650	0.3052	0.7457	0.4093
	*LcGA2ox8*	VIT_206s0004g06790	0.5502	0.8525	0.6454
	*LcGA2ox6*	VIT_205s0077g00520	0.5477	1.3039	0.42
	*LcGA2ox6*	VIT_207s0005g01920	0.5626	1.0286	0.547
Lc/Os	*LcGA2ox1*	OsKitaake01g330800	0.775	0.9395	0.8249
	*LcGA2ox1*	OsKitaake01g077300	0.77	1.2028	0.6402
	*LcGA20ox3*	OsKitaake01g424900	0.351	0.9514	0.3689

## Data Availability

The datasets supporting the conclusions and description of a complete protocol can be found within the manuscript and its additional files. The datasets used and analyzed during the current study are available from the corresponding author on reasonable request.

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
