# Peer review of "Gibberellin Oxidase Gene Family in L. chinense: Genome-Wide Identification and Gene Expression Analysis"

_ijms, 2021, doi:10.3390/ijms22137167_

Round 1
Reviewer 1 Report
The article contains an interesting qualitative study devoted to the analysis of genes belonging to the GA family. However, despite a lot of data, the work cannot be accepted in its present form, as it contains a number of errors and inaccuracies. I hope that the correction of these annoying inaccuracies will make the publication more useful and clear for the readers.
The introduction section is generally informative and well written. However, on line 78, there must be a different word next to the word catalitic, since the meaning of the sentence is not clear.
At the end of the introduction, the research objectives should be clearly stated. However, it contains descriptive lines about what has been done in the work - from 85 to 87, and the meaning of lines 88 - 90 located below is also not clear and has no relation to the goals and objectives, apparently these are fragments from another part?
The Materials and Methods section also requires revision in terms of the description of the experimental object / objects, methods of their cultivation, culture media, since at least three options are mentioned: whole plants, cell culture, protoplasts - all this should be described in separate paragraphs of this section.
In addition, the entire text of the article contains an extremely undesirable "error". One gets the impression that the authors do not know how plant tissues differ from organs, which is extremely important, since the authors did not study any plant tissues in principle. Meanwhile, both in the supplements and in the text of the article, leaves, roots, cotyledons and stems are mysteriously called tissues, while if the authors examined tissues, they would certainly investigate, for example, the epidermis, parenchyma / mesophyll, protoxylema, and so on. This is very important, as it misleads the reader. Of course, the data on tissues would be much more interesting, if only due to the fact that different tissues in plants often have different ploidy, which, in a funny way, is always ignored in molecular biological studies. Apparently, it seems to the authors that a polyploid cell of a developing vessel having several identical chromosomes, according to its ploidy, shows equal expression with a diploid cell in, for example, a meristem. The change in the division process under the influence of stress factors influences even more significantly, when division disorders occur with an additional change in ploidy, which is a well-studied process.
However, since most of the works similar to this study ignore this knowledge, this article will still be interesting since it allows you to compare studies on the principle of the presence of the same error.
Also, the Materials and Methods section does not describe an experiment with the production of transgenic plants and protoplasts. My opinion is that the data of this experiment are not needed in this work and do not make sense in essence, or the description does not allow evaluating this part, then it should be expanded. From the descriptions, I failed to understand the essence of comparing the results of the GFP fluorescence with two genes under the viral constitutive promoter. If the authors believe that there was a certain regulatory sequence in the studied genes, this should be described. What is the essence of comparing these constructs, where is their description, what method of transfection was used? In addition, figure 11 shows protoplasts in which the cell membrane and plastids are fluorescence, additional cytoplasm fluorescence is not obvious, even if, according to the authors, it is present. In addition, it is incorrect to bring one cell when working with unaligned protoplasts, since they often differ in size, the number of chloroplasts / plastids, and in addition, during vivo recording, they will glow without a fluorescent protein ... Moreover, the shooting parameters from the description are not clear to me.
Other remarks.
line 578, table 4 is indicated, but there are primers on it in the appendix, apparently you need to indicate Table 3. Once again, I note that organs and tissues are not the same - check the entire text.
line 407 and the entire next section should either be removed, or why they are described in the article and edits made to the Methods, Figures, Discussion.
Line 416 - it is not clear what prediction are you talking about? What did the authors want to confirm? Correct the goals and objectives in the Introduction section and in the Results, Discussion and beyond.
line 605 - what preference is in question, reformulate what exactly is shown and what does it mean?
line 607
Write about differences in tissues ... you have not analyzed them ... write about organs!
Then there is a statement with the meaning "maybe" - the conclusion should contain a short result, conclusion and perspectives if possible ... The statement in the penultimate sentence could have been written before the article and will remain true after - this is not a result, Correct.
I think the work is big, interesting and it will be better if we eliminate these shortcomings that make it not ready.
Reviewer 2 Report
This manuscript presents data on the family of gibberellin oxidase in Liriodendron. It is a pretty accurate work, with a lot of information. As an initial comment, however, I believe that it is important to finalize the work more by clearly expressing the overall purpose.
The abstract is well written even if the motivation of the work is restricted to the first 2-3 lines of text. Moreover, the beginning of the abstract focuses more on the importance of studying certain gene flows for market issues, while at the end of the abstract reference is made to resistance to environmental stress conditions. It is therefore necessary to clarify the purpose of the manuscript.
I advise authors to write the name of enzymes in a complete way, at least the first time they are mentioned, as they did for gibberellin oxidase.
The first paragraph of the introduction describes the three fundamental steps leading to the synthesis of gibberellins, while the second paragraph deals specifically with the third step of synthesis of gibberellins. However, there is no reason why the authors have decided to focus only on the third step of gibberellin synthesis.
The motivation to study gibberellin oxidase is described in the third paragraph of the introduction, when the authors introduce the plant species of interest. I believe, however, that this explanatory statement should be brought forward in the text.
The title of the manuscript explicitly talks about abiotic stress but in the whole introduction there is no mention of the effects of stress, even just to explain why the word "abiotic stress" is present in the title. I believe that the introduction lacks an important part relating to stress.
Part of the results is basically the presentation of bioinformatics analyses with the aim of describing the gene similarity between different plant organisms, as well as the structure of the genes and proteins they encode and the use of codons. It is a very comprehensive section; I have nothing to point out.
The link between the family of gibberellin oxidase and the response to abiotic stress arises from the identification of particular gene elements in the promoters of genes coding gibberellin oxidase. Among the various modules present in the promoters there are elements that indicate gene expression in relation to stressful conditions.
There is, however, one aspect that I did not fully understand in this work, and that is the part relating to the impact of stress conditions on the expression of gibberellin oxidase. The authors tested some stressful conditions, including PEG, NaCl, etc. and then analyzed the effects on the expression of gibberellin oxidase. At first glance, however, it seems something slightly superficial, there is no real deepening of this part. For example, what does this data mean? What leads this different expression of 2-3 gibberellin oxidase genes under stressful conditions? What is the feedback? What is the effect? What is the advantage that this different expression can provide to plants? At first glance the stress aspect is very secondary to the discussion of phylogenetic analysis or the gene structure of gibberellin oxidase. And even in the discussion the section on abiotic stress is discussed quickly to confirm that it is not the main interest of authors.
The part about the expression of gibberellin oxidase fused with GFP has little value; it is a confirmation of something already expected and adds nothing more, I did not find the meaning of this test.
In conclusion, I would like to return to the first point I mentioned, namely the abstract where, in the first lines of text, it is indicated that one of the reasons for studying this species is the demand of the market. All this, however, is completely missing in the manuscript which instead is 90% focused on the gene structure of gibberellin oxidase. I believe that the primary interest of the authors is related to this gene family, its phylogenetic relevance, the way by which genes for gibberellin oxidase are expressed. I also believe that the interest in abiotic stress is secondary to the predominant part and therefore I believe that having included the word "abiotic stress" in the title is confusing and generates an expectation that is not found in the text. I would honestly suggest that authors do not talk about abiotic stress, at least in the title, and that they stress more about work on gene organization. Even the plant species of interest is ultimately treated very superficially, there is no final return to the plant itself, nor even something that is related to the economic demands of the market. I think it is necessary to make more emphasis on the importance of this knowledge for the species analyzed.
Round 2
Reviewer 1 Report
We note that the authors have done some work to improve the provided manuscript. In general, the article can be accepted for publication in the International Journal of Molecular Sciences. However, it is nevertheless necessary to make minor corrections to the text of manuscript.
Line 61 What does «oranges» mean in this sentence?
Line 78-79 «Gibberellin oxidase is considered to be an important catalytic and synthase…» - This is unclear. Reframe the sentence. Because, the word “catalytic” is an adjective.
Line 335 “Oranges” What it is?
Line 355 “Different oranges”?
Line 398 “Orange parts”?
Line 461 Oranges
Line 464 young oranges
Line 467 orange stage
Line 486 different oranges
Line 551 orange transcript
Line 567 orange
Line 586 different oranges
Dear Hu Lingfeng!
"Organ" is not "orange" (Citrus sinense)!
It's awfully! Please correct.
